# Preparation and Performance Study of Graphene Oxide Doped Gallate Epoxy Coatings

**DOI:** 10.3390/ma18153536

**Published:** 2025-07-28

**Authors:** Junhua Liu, Ying Wu, Yu Yan, Fei Wang, Guangchao Zhang, Ling Zeng, Yin Ma, Yuchun Li

**Affiliations:** 1China National Chemical Construction Investment Group Co., Ltd., Beijing 100041, China; m15386400818@163.com (J.L.); 15388014390@163.com (Y.Y.); wangfei202507@163.com (F.W.); zhanggc1229@163.com (G.Z.); 2School of Chemistry and Pharmaceutical Engineering, Changsha University of Science and Technology, Changsha 410114, China; wying_989464@163.com (Y.W.); 22209031821@csust.edu.cn (Y.M.); 3School of Civil and Environmental Engineering, Changsha University of Science and Technology, Changsha 410114, China; zlbingqing3@126.com

**Keywords:** methyl gallate, graphene oxide, anti-rust coating, scanning vibration electrode technique

## Abstract

Coatings that are tolerant of poor surface preparation are often used for rapid, real-time maintenance of aging steel surfaces. In this study, a modified epoxy (EP) anti-rust coating was proposed, utilizing methyl gallate (MG) as a rust conversion agent, graphene oxide (GO) as an active functional material, and epoxy resin as the film-forming material. The anti-rust mechanism was investigated using potentiodynamic polarization (PDP), electrochemical impedance spectroscopy (EIS), scanning electron microscopy (SEM), laser scanning confocal microscopy (LSCM), and the scanning vibration electrode technique (SVET). The results demonstrated that over a period of 21 days, the impedance of the coating increases while the corrosion current density decreases with prolonged soaking time. The coating exhibited a maximum impedance of 2259 kΩ, and a lower corrosion current density of 8.316 × 10^−3^ A/m^2^, which demonstrated a three-order magnitude reduction compared to the corrosion current density observed in mild steel without coating. LSCM demonstrated that MG can not only penetrate the tiny gap between the rust particles, but also effectively convert harmful rust into a complex. SVET showed a much more uniform current density distribution in the micro-zones of mild steel with the anti-rust coating compared to uncoated mild steel, indicating that the presence of GO not only enhanced the electrical conductivity of the coating, but also improved the structure of the coating, which contributed to the high performance of the modified epoxy anti-rust coating. This work highlights the potential application of anti-rust coating in the protection of metal structures in coastal engineering.

## 1. Introduction

In complex marine environments, mild steel is typically subjected to a combination attack of seawater immersion and exposure to salty atmospheric conditions, which significantly reduces its service life. To achieve effective anti-corrosion with traditional coatings, meticulous surface descaling treatment is imperative. However, in practice, rust removal procedures such as sandblasting are ineffective. Such pretreatment fails to meet the stringent Sa 2.5 standard requirements (ISO8501-1 [1]) and compromises coating quality [2,3,4]. Therefore, it is essential to develop a tolerant coating for poorly prepared surfaces that can effectively mitigate rust formation and enhance construction operations.

Corrosion-resistant coatings are applied directly to inadequately prepared surfaces with minimal pre-treatment, such as substrate descaling [5,6,7]. Currently, tannic acid and phosphoric acid are the predominant rust inhibitors used in rust conversion agents, effectively enhancing the overall corrosion resistance of the coating [5,8,9,10]. In a study conducted by Liu et al., tannic acid was selected as the anti-rust conversion agent, supplemented with D-limonene and nano-ZrO_2_. The electrochemical impedance results demonstrated that the impedance of the optimal coating, treated with the anti-rust conversion method and additives, reached 10^7^ Ω·cm^2^, exhibiting an increase of three orders of magnitude compared to that of the original coating [11,12,13].

Shicheng et al. found that the composite coating exhibited excellent long-term corrosion resistance due to the maze effect of montmorillonite (MMT) and the synergistic effects of Ce^3+^ and TA corrosion inhibitors, thereby demonstrating significant potential application in various fields [14,15,16,17]. The tannic acid and phosphoric acid conversion system presents several challenges, including exorbitant raw material costs, environmental unfriendliness, and subpar adhesion of the coating film [18,19]. Furthermore, the efficacy of tannic acid in rust conversion is closely linked to its concentration; excessive or insufficient amounts yield unfavorable outcomes. An insufficient amount results in incomplete reaction with rust, while excessive addition compromises coating stability [20,21,22,23].

Despite the limited durability of organic coatings, epoxy resin-based coatings are still recognized for their protective abilities and their effectiveness in treating the rust layer. Gallate can exhibit cost-effectiveness, environmental friendliness, and facile synthesis. The presence of the hydroxyl group in gallate not only enhances its miscibility with epoxy (EP) resin but also enables its involvement as an intermediate [24,25,26,27]. Graphene (G) is the thinnest two-dimensional carbon material, and its unique nanostructure renders it impermeable to any atom and molecule under ambient conditions. Therefore, hydrophilic graphene oxide (GO) nanosheets have interlayer channels that act as molecular sieves, exhibit superior dispersion, and can serve as an exceptional conductive functional filler in anti-corrosion coatings [28,29,30,31].

Based on the advantages of GO active functional materials, we have added GO components to epoxy-modified MGs. The article focused on the preparation of modified epoxy with methyl gallate (MG) and GO for the common mild steel Q355B. The modified anti-rust coating improved the integrity and shielding of the coating and significantly enhanced the corrosion resistance of the mild steel structure subjected to long-term corrosion in coastal engineering.

## 2. Materials and Methods

### 2.1. Materials

The water-based epoxy adhesive system (JH-5560, Hangzhou Wuhuigang Adhesive Co., Ltd., Hangzhou, China) consists of two components: Part A, an epoxy emulsion (bisphenol A type, resin content 50 ± 2 wt%, epoxy value 200~220 g/mol); and Part B, a waterborne polyamide curing agent (amine value 180~190 mg KOH/g). The following chemicals were used in this study: silane coupling agent (KH-550, γ-aminopropyltriethoxysilane, purity ≥ 98%), methanol (HPLC grade, ≥99.9%), gallic acid (GA, 3,4,5-trihydroxybenzoic acid, purity ≥ 99%, p-toluene sulfonic acid (TsOH, purity ≥ 98.5%), GO, dispersant (DK-006, polyacrylate copolymer), and leveling agent (RM-2020, polysiloxane-based). All chemicals were of analytical grade and purchased from Shanghai Macklin Biochemical Co., Ltd. (Shanghai, China).

### 2.2. Preparation of MG

The reactions were performed in 250 mL three-necked flasks containing 150 mL of absolute methanol. To this solvent, GA (5.0 g, 29.4 mmol) and TsOH (0.25 g, 1.47 mmol, 5 mol% relative to GA) were added as reactants and catalysts, respectively. The mixture was first stirred in a water bath at 55 °C (300 rpm) for 1 h to ensure uniformity, then heated to 100 °C and held under reflux for 4 h. After complete dissolution of the solid, the system was cooled to room temperature (25 ± 2 °C), sealed under a nitrogen atmosphere, and aged for 24 h. The resulting product was freeze-dried to obtain MG composites.

### 2.3. Preparation of Anti-Rust Coating

To achieve the ideal dispersion effect, 3 wt% of GO was treated under ultrasonic conditions with a dispersant (DK-006, a polymer-type dispersant with a specific molecular structure and functional groups, provided by Guangdong Zilibon Chemical Co., Ltd., Dongguan, China) until a uniform dispersion was obtained. It is soluble in a variety of organic solvents, but insoluble in water and adsorbs firmly on the surface of pigments or fillers through anchoring groups. Based on experimental analysis, the ideal control condition for ultrasonic dispersion is 20 kHz for 40 min.

At the same time, its solvated segments extend in systems such as epoxy coatings, forming an effective steric hindrance to prevent the mutual aggregation of pigment or filler particles, thus achieving a good dispersing effect. Subsequently, the mixture was supplemented with 5 wt% MG and epoxy resin for thorough blending and stirring, followed by ultrasonication for a duration of 30 min. Finally, 50 wt% curing agent was swiftly incorporated into the mixture, followed by thorough stirring to obtain an anti-rust coating, as shown in Figure 1. The arrows in the figure represent the direction of the preparation process, graphene oxide is represented by a thin mesh symbol, MG is represented by a circle, the small black dots and yellow dots in the beaker represent EP, and the SVET plot at the end of the process represents the electrochemical morphology of the coating micro-zone.

### 2.4. Preparation of Coating Film

Q355B (GB/T 1591-2018) [32] carbon steel was used in measurement, with dimensions of 50.0 mm × 20.0 mm × 2.0 mm. Prior to experimentation, the material underwent a thorough cleaning process involving acetone and ethanol solvents. Subsequently, it was immersed in a simulated seawater solution (the preparation method of the simulated seawater solution is to weigh a certain amount of NaCl, MgCl_2_, Na_2_SO_4_, CaCl_2_, KCl, NaHCO_3_, NaBr, H_3_BO_3_, SrCl_2_, and NaF, add them to an appropriate amount of distilled water in turn, stir until completely dissolved, and then make up to volume in a 1 L volumetric flask) for a total duration of 6 h, followed by intermittent exposure to the natural atmospheric environment for 12 h. After corrosion occurred over a period of 7 days, a complete and uniform rust layer formed on the surface. To remove any loose rust particles while retaining firmly attached rust, gentle sandpaper polishing was performed on the corroded substrate [33]. Finally, the surface was coated with MG-modified anti-rust coating; the thickness of the coating was controlled at 125 ± 25 μm, followed by thermal treatment at 60 °C for 12 h to ensure complete curing. The cured samples were subsequently tested as described.

### 2.5. SEM/LSCM/XRD Analysis

The morphology and composition of the coating and corrosion products were analyzed using a FEG250 scanning electron microscope (SEM, FEI Company, Hillsboro, OR, USA). Laser Scanning Confocal Microscope (LSCM, KEYENCE Company, Osaka, Japan) is an effective technique to provide three-dimensional (3D) surface topography and was employed to study oxidation behavior in this work [34,35,36,37]. The surface topography and oxide thickness were measured using a 3D LSCM (VK-150K). X-ray diffraction (XRD, Bruker D8 ADVANCE, Bruker, Billerica, MA, USA) was used to analyze the phase composition of corrosion products under conditions of 40 kV voltage and 30  mA current.

### 2.6. AC Impedance and Polarization Analysis

The AC impedance spectrum and potentiodynamic polarization curve of the anti-rust coating were measured using a Chenhua CHI660E electrochemical workstation (Shanghai Chenhua Instrument Co., Ltd., Shanghai, China). A three-electrode working system was employed, with a platinum sheet as the auxiliary electrode, a saturated calomel electrode (SCE) as the reference electrode (all potentials in this work are relative to the SCE), and simulated seawater solution as the corrosion medium. The components of the simulated seawater solution are some inorganic salts and trace elements, and the specific components are listed in Section 2.4. It complies with the provisions of ASTM D1141 [38]. The test area was set at 1 cm^2^, while the scanning frequency for the AC impedance spectrum ranged from 10^−2^ to 10^5^ Hz with an amplitude of 10 mV [39,40,41]. The open-circuit potential (OCP) was monitored using a CHI660E electrochemical workstation, and OCP vs. time curves were recorded for 120 s per measurement until a steady-state value was achieved. The potentiodynamic polarization scan began at OCP and ended at −250 or 250 mV (vs. E_corr_) at a scan rate of 1 mV/s. The OCP lasted at least half an hour until the potential stabilized, followed by potentiodynamic polarization. Three valid experiments were performed for the AC impedance tests and the potentiodynamic polarization test under the same conditions, and the average of the three experiments was obtained for the analysis.

### 2.7. SVET Measurements

One of the most crucial methods for analyzing coatings involves utilizing a micro-tip to investigate the current distribution over a corroding surface or substrate [42,43,44,45,46]. The primary outcome obtained through the scanning vibration electrode technique (SVET) is a potential map measured at the micron scale in corrosion systems. The SVET test was carried out in simulated seawater solution using the Princeton VersaSCAN micro area electrochemical comprehensive test system produced by AMETEK, Berwyn, PA, USA.

These maps reveal both reduction reactions occurring on the cathode and oxidation reactions taking place on the anode. When immersed in a conductive electrolyte, the vibrations of the microelectrode allow the probe to measure the potential difference at two positions along the vibration direction, which allows for the determination of the ionic current map on the scanned surface. For instance, Nardeli et al. demonstrated SVET maps depicting unmodified and tannin-modified polyurethane coatings while elucidating their healing process [47,48].

The vibrating electrode was made of platinum–iridium and covered with a polymer, leaving only an uncovered tip with a diameter of ~10 μm; platinum black was deposited on the probe tip. The distance of the tip to the surface was kept at 100 μm. The vibration frequency was set to 80 Hz, and the peak-to-peak amplitude was 30 μm. During the measurements, the probe vibrated in the vertical (Z) direction at the frequency mentioned above, while along the X and Y directions, it moved stepwise at a speed of less than 300 μm/s in the horizontal plane. The scanning data points were acquired 100 ± 2 μm above the sample surface.

## 3. Results

### 3.1. SEM Analysis

Figure 2 shows the surface morphology and elements mapping of Q355B steel with and without the anti-rust coating. The scanning electron microscope image of Q355B without coating is depicted in Figure 2a. The characteristic large-scale pits observed on the coating surface suggest three concurrent degradation pathways: (i) electrochemical corrosion, evidenced by iron oxyhydroxide/hematite (FeOOH/Fe_2_O_3_) product layers; (ii) interfacial delamination between the coating and substrate; and (iii) multifaceted aging mechanisms. Specifically, the aging process involves the synergistic effect between thermal oxidative degradation, solvent osmotic swelling, and other non-corrosive factors. Nevertheless, owing to the presence of the anti-rust coating in Figure 2b, a significant portion of the coated surface remains relatively intact and smooth, thereby effectively impeding the penetration of corrosive media between the coating and substrate. From these images, it is evident that the coating exhibited relative flatness and uniform distribution.

The element composition of the corresponding area in the SEM image was tested using EDS analysis. Figure 2c shows the EDS spectra of Q355B with the anti-rust coating. The key elements include C, O, Na, and Cl on the sample surface. The EDS software (EDS detector dataset, version 3.2.1) compares the characteristic peak intensities of the elements (e.g., the Kα peak of C ≈ 0.28 keV) with the database of the standard samples to obtain the order of the contents of C, O, Na, and Cl from high to low. The atomic ratio of these four elements is C:O:Na:Cl = 98.24:1.34:0.01:0.35. It can be seen that the coating was mainly composed of C and O, primarily provided by GO and MG. The absence of Fe in the surface EDS analysis (detection depth ~3 μm) suggests effective local coverage, although the 150 μm coating thickness requires cross-sectional characterization to fully assess interface integrity. The presence of Na and Cl was due to the presence of a large amount of sodium chloride in the simulated seawater solution. These results suggest that the GO powder was evenly mixed with MG in the epoxy matrix, and that the anti-rust coating remained stable in the corrosive environment without significant degradation, which is consistent with the results of electrochemical testing and analysis.

### 3.2. LCSM Analysis

Figure 3 shows the 3D surface topography of Q355B with and without anti-rust coating obtained by LSCM. The surface of the rust layer changed from the original porous shape to a flat and uniform film morphology, and the original loose corrosion products may have reacted with the MG in the coating and were subsequently covered by the film. SEM observations suggest that the MG formulation may penetrate interparticle voids in the surface rust layers; however, verifying its distribution within the 150 μm coating requires cross-sectional analysis to resolve potential leveling effects. The coordination reaction between iron ions and MG may convert the harmful rust into a complex or chelate, which provides protective effects on the substrate and promotes the appearance of a flat film. GO further enhances the homogeneity and conductivity of the coating. Additionally, LCSM can indirectly acquire membrane thickness information through 3D imaging capabilities. The surface topography of Q355B with anti-rust coating in Figure 3b is smoother than that of Q355B without coating in Figure 3a; the thickness of surface topography was 27.056 μm and 1.836 μm, corresponding to Q355B without and with anti-rust coating, respectively, demonstrating the strong protective effect of the anti-rust coating through the significant reduction of corrosion products.

### 3.3. Electrochemical Analysis

#### 3.3.1. Tafel Polarization Curve Analysis

Figure 4a shows the potentiodynamic polarization (PDP) curves of Q355B steel coated with anti-rust coating after soaking in simulated seawater solution for 1 day, 7 days, 14 days, and 21 days, while Figure 4b presents the trends of open circuit potential (OCP) versus time. The PDP test was repeated three times for each condition. Q355B steel treated with anti-rust coating had an OCP of −0.622 V (SCE) and a corrosion current density (i_corr_) of 2.190 A/m^2^ after soaking in simulated seawater solution for 1 day. After soaking for 7 days, the OCP was −0.584 V and i_corr_ was 0.023 A/m^2^. After soaking for 14 days, the OCP was −0.556 V and i_corr_ was 0.015 A/m^2^. Additionally, after soaking for 21 days, the OCP was −0.539 V and i_corr_ was 0.008 A/m^2^. With prolonged immersion, the OCP value of the coated sample shifted positively from −0.622 V to −0.539 V, while the corrosion current density decreased with the increase of soaking time. The OCP test was repeated three times for each condition. After soaking for 21 days, the corrosion current density of Q355B steel with anti-rust coating decreased by three orders of magnitude compared to uncoated Q355B. In comparison, the OCP of Q355B steel without coating was −0.685 V, and the i_corr_ was 25.84 A/m^2^. Figure 4c shows the corrosion current density vs. time for coated vs. uncoated. The coated sample soaking for 21 days exhibited the lowest corrosion current density and highest OCP, indicating optimal corrosion resistance. After 21 days, OCP stabilized at −540 mV. Table 1 summarizes the electrochemical parameters of PDP and OCP measurements. The anode Tafel slope (b_a_) was 4.445 when the coating was soaked for 21 days, which indicated the lowest polarization level in comparison with other time periods. This phenomenon may be attributed to the formation of stable iron–MG complexes via coordination between phenolic hydroxyl groups and iron ions, which inhibits the further formation of rust. The observed bubble-mediated coating growth originates from oxygen evolution (O_2_ + 2H_2_O + 4e^−^ → 4OH^−^) during cathodic corrosion. A comparative experiment was also carried out for Q355B steel with EP/GO coating, in which the OCP was −0.605 V, and the i_corr_ was 12.41 A/m^2^. This mechanism involves chemical chelation, where methyl gallate (MG) phenolic hydroxyl groups coordinate with Fe^2+^ to form stable Fe^3+^-catechol complexes. GO can improve anti-corrosion properties due to its layered structure and a large number of oxygen-containing groups, which improves the compositional distribution of the coating.

#### 3.3.2. Electrochemical Impedance Analysis (EIS)

The EIS data were analyzed following Mansfield’s approach, wherein the measured impedance spectra (Figure 5) were validated via Kramers–Kronig transforms prior to equivalent circuit modeling. The optimal circuit topology (R in Zview) was selected based on the following criteria: (i) x^2^ < 1 × 10^−3^; (ii) <5% parameter covariance. Charge transfer resistance and mass transport limitations were quantitatively resolved through this protocol [49]. A larger capacitance arc diameter corresponds to greater charge/mass transfer resistance and enhanced corrosion resistance, whereas a smaller arc indicates reduced resistance. Figure 5 shows the equivalent circuit diagram of impedance transformation for (a) a coated system and (b) an uncoated system. As depicted in this figure, Rs represents the solution resistance, Qc denotes the interface capacitance between the electrode and the solution, and Rp corresponds to the coating resistance. For coated systems, Qdl and Rct are used in the equivalent circuit to represent the coating; Qdl reflects the interface capacitance between the electrode and coating, while Rct indicates the charge-transfer resistance.

Through data normalization based on equivalent circuit fitting, Figure 6a shows the tendency of electric components. As evident from the data, the solution resistance (Rs) ranged between 23.4 kΩ and 23.6 kΩ, with little temporal fluctuation. The Rct values of the coating after soaking for 1 day, 7 days, 14 days, and 21 days were 73.2 kΩ, 108.6 kΩ, 1984 kΩ, and 2259 kΩ, respectively. The observed impedance peaks at 1 day and 21 days coincided temporally with corrosion product accumulation. Although this correlation may suggest localized ion-migration blocking, direct mechanistic confirmation requires further interfacial characterization. The <5% R_s_-Rct covariance confirmed the model’s robustness against solution resistance (R_s_) variations below the 5% interference threshold. It is noteworthy that the average value of the solution resistance over the four time periods was used in data processing; therefore, it has little influence on the subsequent tests. Figure 6b represents the Nyquist plots of Q355B steel treated with anti-rust coating immersed in simulated seawater solution for different times. It is clear that the Nyquist plots of each group form a semicircle, and as the soaking time increased, the radius of the semicircle also increased continuously, consistent with the Rct tendency in Figure 6a. After 21 days of immersion, the Nyquist semicircle exhibited the maximum diameter (Rct = 2259 ± 0.5 kΩ·cm^2^), corresponding to a 0.38% decrease in i_corr_ compared to day 1. This temporal improvement aligns with typical polymer coating stabilization processes, though long-term durability requires further validation. This is because, as the soaking time of the coating in the simulated seawater solution increases, the phenolic hydroxyl groups contained in MG can further combine with iron ions to form a series of stable complexes, thereby inhibiting the further generation of rust. Therefore, the resistance of the coating surface to solution erosion continued to increase, and the Rct value also increased continuously, which was consistent with the Tafel analysis results. Figure 6c,d shows the Bode plots of Q355B steel treated with anti-rust coating immersed in simulated seawater solution for different times. A similar pattern can be observed: impedance decreased while phase angle fluctuations increased with increasing frequency, indicating that the structure of the bilayer on the surface of Q355B matrix underwent a complex transformation and enhanced the corrosion resistance of the material.

## 4. Discussion

The SVET was used to evaluate the local electrochemical corrosion reaction of metals beneath the coating in the simulated seawater solution. Figure 7 displays the SVET maps of potential distribution under a 1.5 V signal applied to sample surfaces, including (a) coated samples and (b) uncoated samples. The potential scale magnitude ratio between Figure 7a,b is 1:40. To enable direct comparison, Figure 7c integrates both datasets through numerical scaling normalization. In Figure 7c, black/gray regions represent coated areas. Notably, the uncoated sample exhibits significantly greater potential fluctuations (Figure 7b) compared to the coated system (Figure 7a). It can also be found that Figure 7a indicates cathodic/anodic reaction potential fluctuations in the range of −0.032~0.023 mV, while Figure 7b indicates cathodic/anodic reaction potential fluctuations in the range of −12~4.41 mV.

The obtained potential signals can be converted to the local current density expressed with Ohm’s law [43], as shown in Equation (1), where Δϕ is the electric potential drop, *k* is the electrolyte conductivity, and *d* is the vibration amplitude. During the SVET measurements, the peak-to-peak amplitude was set to 30 μm, and the electrolyte conductivity of the simulated seawater solution was 2450 μS/cm; the conversion factor between J and Δϕ was 816.7.(1)$$J=-\Delta \phi \cdot \frac{k}{d}({\mu A/\mathrm{cm}}^{2})$$

Figure 8 illustrates the distribution of micro-zone current density along the z-axis with respect to the x-axis direction in the SVET test, under a 1.5 V signal applied to the sample surfaces, including (a) coated samples and (b) uncoated samples for different immersion times. The figure provides a comprehensive comparative analysis of the 3D data in a two-dimensional manner, each data point in the figure represents the maximum value of the micro-zone current density in the y-axis direction. The distribution along the x-axis in the figure represents one test micro-area every 300 μm, resulting in 11 test points and 10 segments over a 3 mm range, corresponding to the 11 data points shown; additionally, 100 test micro-areas are distributed across the xy-plane. The black line in the figure represents the moment when the voltage is first applied to the sample, corresponding to 0 s in the legend; the red line represents half an hour after the voltage is applied to the sample, corresponding to 1800 s in the legend; and the blue line represents one hour after the voltage is applied to the sample, corresponding to 3600 s in the legend.

Notably, the micro-zone current density changed as the test progressed. As shown in Figure 8a, at 0 s (black line), there was a very small amount of micro-zone current density with minimal fluctuation, and the average current density was 8.8 μA/cm^2^; at 1800 s, the micro-zone current density increased slightly with a mean value of 11.0 μA/cm^2^; at 3600 s, the increase in micro-zone current density was more significant, and the degree of fluctuation also increased, with an average value of 21.9 μA/cm^2^. However, Figure 8b shows a much larger micro-zone current density; even at 0 s, its average value was 22.9 μA/cm^2^, which is larger than that of 3600 s in Figure 8a. The uncertainty in the micro-zone test data for each condition in Figure 8a suggests that there are very subtle differences in electrochemical performance between the micro-zones, and the data all fluctuate over a small range. At 1800 s, Figure 8b shows that the micro-zone current density increased more significantly and also fluctuated to a greater extent, reaching an average value of 169.3 μA/cm^2^, which was more than 15 times larger than that of Figure 8a. At the time point of 3600 s, Figure 8b showed that the micro-zone current density increased significantly and fluctuated to a large extent, reaching an average value of 706.8 μA/cm^2^, which was more than 30 times larger than that of Figure 8a. Comparison of Figure 8 shows that the anti-rust coating exhibited very low micro-zone current density, and the relative fluctuation of the current density on each micro-zone in the range of 3 × 3 mm was very small, providing very strong protection for the specimen. This indicates that the presence of GO and MG may not only enhance the corrosion resistance of the specimen surface, but also improve the compositional distribution of the material surface.

To further investigate the nature of these corrosion products, X-ray diffraction analysis was conducted on Q355B steel with and without anti-rust coating, and the test results are depicted in Figure 9. As depicted in Figure 9, prior to the application of the anti-rust coating, the substrate surface primarily consisted of Fe, γ-FeOOH, Fe_2_O_3_, and Fe_3_O_4_. In addition to iron, which is the main ingredient, the predominant constituent on the surface of the sample was γ-FeOOH. It can be observed from the XRD spectrum that the peak exhibits a distinct sharpness with high intensity. The attenuated XRD peak intensities (Cu Kα, λ = 0.154056 nm) after coating application reflect the technique’s shallow penetration depth (~20 μm vs. coating thickness = 125 ± 25 μm). This confirms predominant detection of the coating matrix containing dispersed corrosion products, while substrate signals are effectively masked. The sample surface with anti-rust coating consists of various amorphous ferric oxide/ferric hydroxide species, and a significant increase in the content of Fe_3_O_4_ indicates that the GO in the coating oxidizes γ-FeOOH, resulting in a change in the properties of the coating in favor of corrosion prevention. By comparing the results, it can be inferred that the reaction between the rust and the anti-rust conversion agent effectively inhibits corrosion occurrence.

## 5. Conclusions

An anti-rust coating was successfully prepared using MG as a rust conversion agent, GO as an active functional material, and epoxy resin as a film-forming substance. The results showed that within 21 days, with increasing soaking time in simulated seawater solution, the impedance of the coating increased, and the corrosion current density decreased. The maximum impedance of the coating reached 2259 kΩ, and the corrosion current density reached 8.316 × 10^−3^ A/m^2^, which was three orders of magnitude lower than the corrosion current density of Q355B steel without an anti-rust coating. This was because as the immersion time of the coating in simulated seawater solution increased, the phenolic hydroxyl groups contained in MG could further combine with iron ions to form a series of stable complexes, thereby inhibiting the further generation of rust. Surface topography analysis via SEM, EDS, and LSCM suggests potential MG infiltration at rust particle boundaries. SVET showed a very uniform distribution of potential and current density for Q355B steel with anti-rust coating in comparison with that of Q355B steel without coating and with only epoxy, indicating that the presence of GO not only enhanced the electrical conductivity of the coating, but also improved the structure of the coating. This work demonstrates the potential application of anti-rust coating in the field of protection for mild steel in coastal engineering.

## Figures and Tables

**Figure 1 materials-18-03536-f001:**
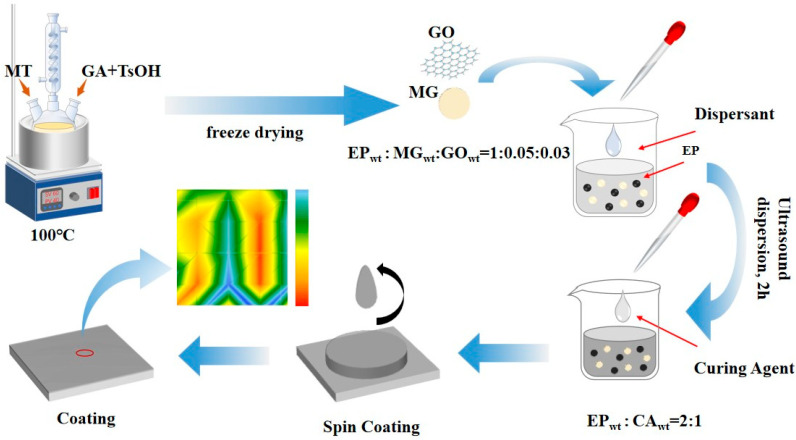
Fabrication process of anti-rust coating.

**Figure 2 materials-18-03536-f002:**
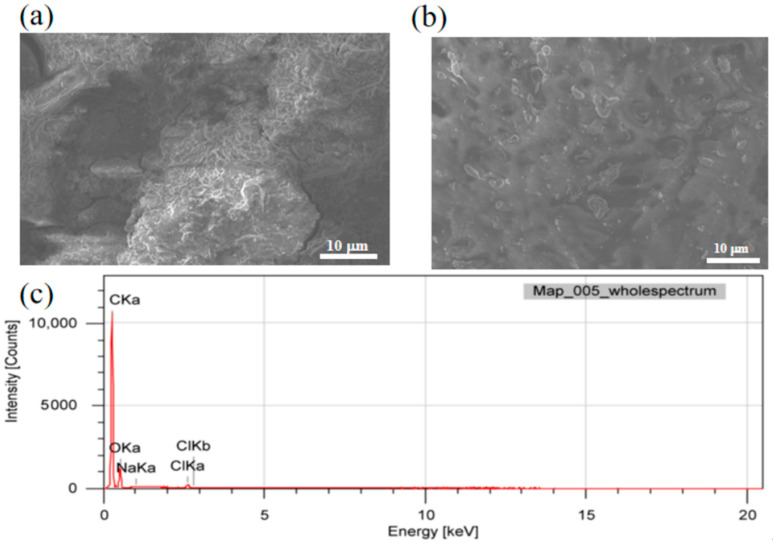
SEM morphology of Q355B steel without and with anti-rust coating: (**a**) SEM of Q355B steel; (**b**) SEM of Q355B steel with anti-rust coating; (**c**) EDS spectrum of the entire area in (**b**).

**Figure 3 materials-18-03536-f003:**
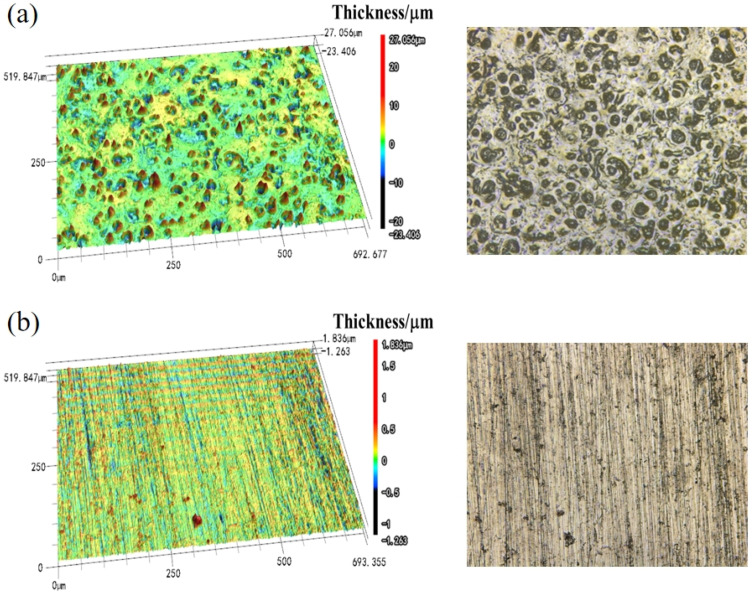
(**a**) LSCM of Q355B steel; (**b**) LSCM of Q355B steel with anti-rust coating.

**Figure 4 materials-18-03536-f004:**
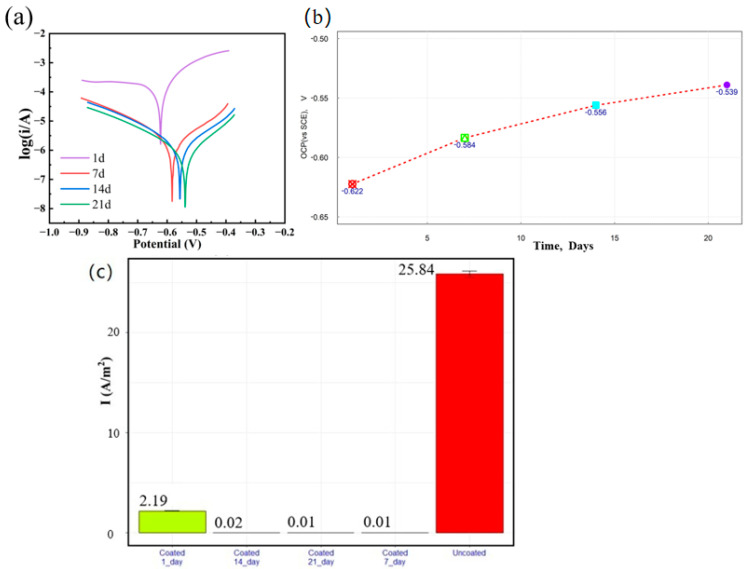
(**a**) Potentiodynamic polarization curves of Q355B steel treated with anti-rust coating immersed in simulated seawater solution for different times; (**b**) Plot of OCP versus time; (**c**) Corrosion current density vs. time for coated vs. uncoated.

**Figure 5 materials-18-03536-f005:**
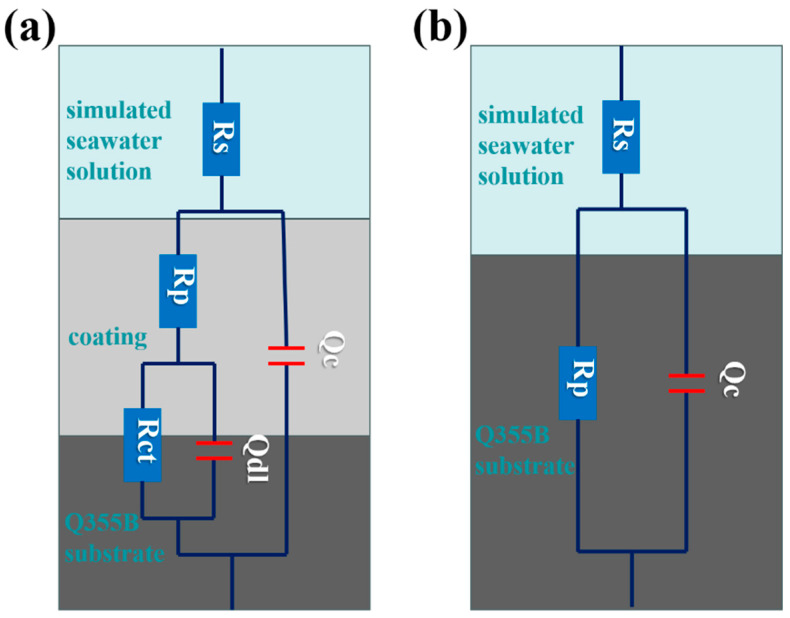
EIS equivalent circuit diagram of Q355B: (**a**) with coating and (**b**) without anti-rust coating in simulated seawater solution.

**Figure 6 materials-18-03536-f006:**
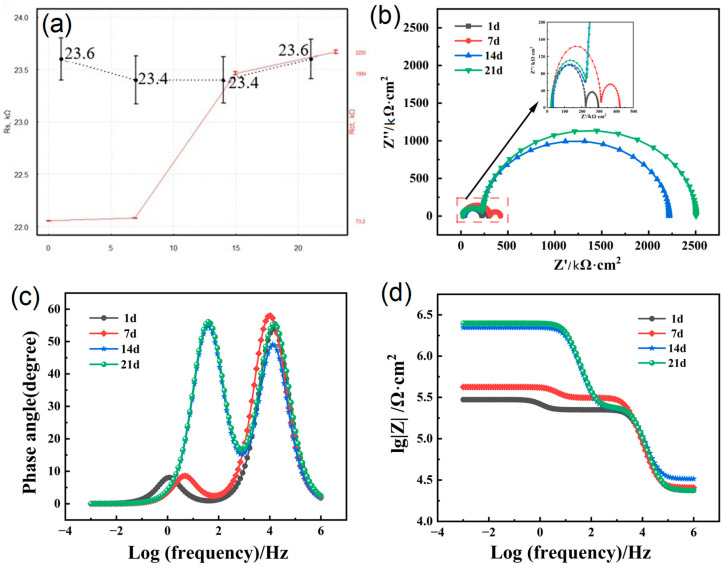
(**a**) Analysis of tendency of electrochemical components by fitting the equivalent circuit; (**b**) Nyquist plots of fitted data for Q355B steel with anti-rust coating immersed in simulated seawater solution for different times; (**c**,**d**) Bode plots of fitted data for Q355B steel with anti-rust coating immersed in simulated seawater solution for different times.

**Figure 7 materials-18-03536-f007:**
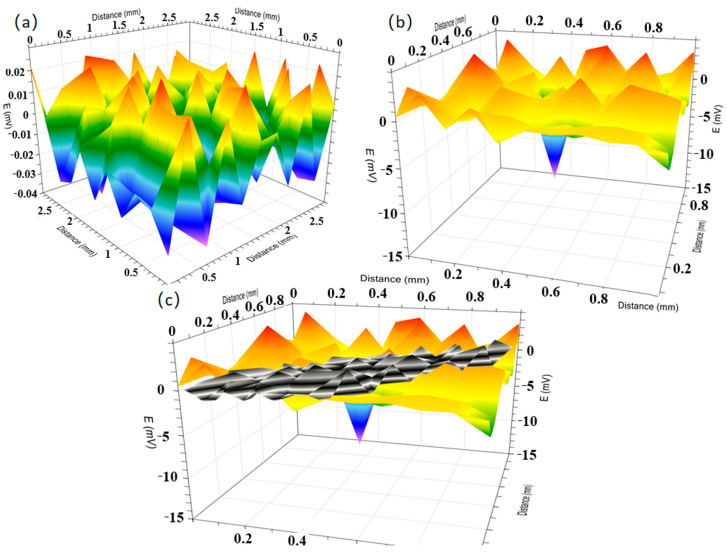
SVET potential maps of Q355B steel: (**a**) with anti-rust coating; (**b**) without coating in simulated seawater solution. (**c**) comparison of figure (**a**) with black color and (**b**) with yellow color at the same potential scale.

**Figure 8 materials-18-03536-f008:**
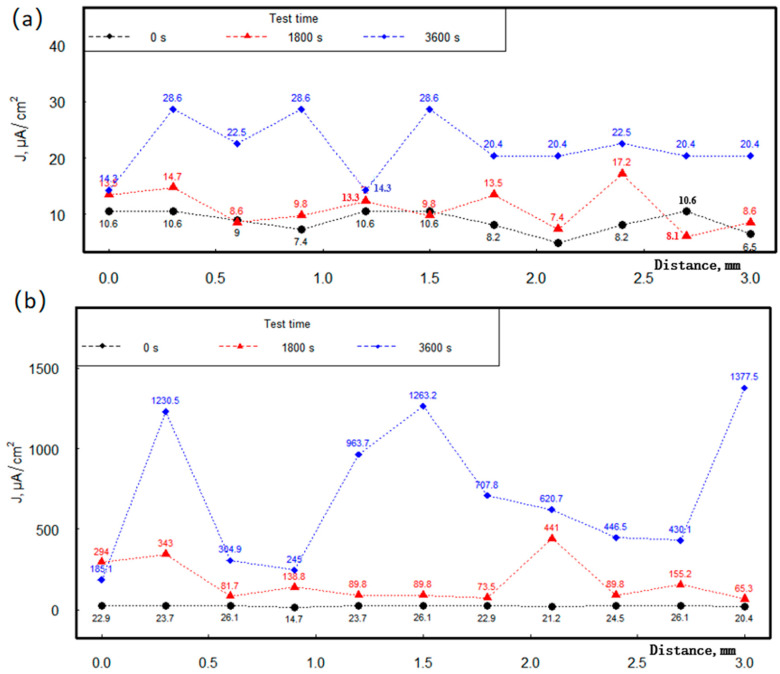
Data analysis comparison diagram of SVET test of Q355B sample: (**a**) with anti-rust coating; (**b**) without coating for different times in simulated seawater solution.

**Figure 9 materials-18-03536-f009:**
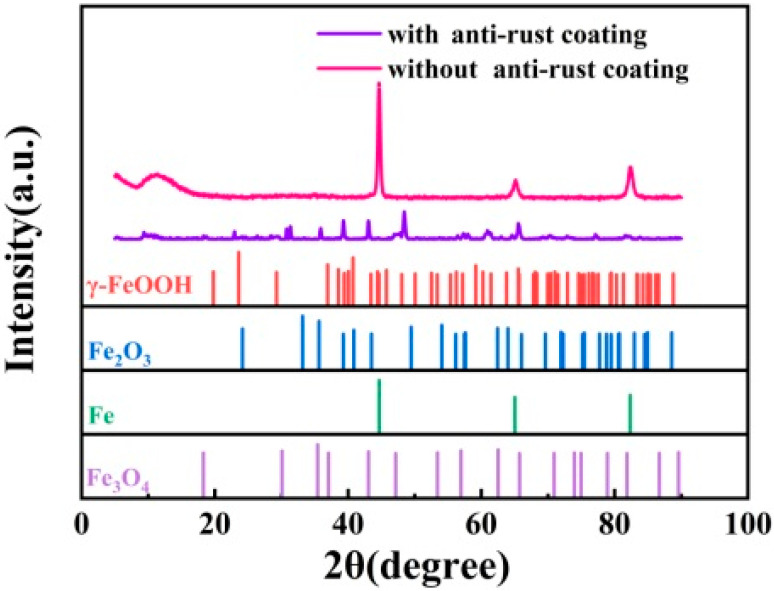
XRD spectrum of Q355B steel with and without anti-rust coating.

**Table 1 materials-18-03536-t001:** Corrosion current density and OCP of Q355B with anti-rust coating after soaking in simulated seawater solution for one day, seven days, fourteen days, and twenty-one days, respectively.

Time, Days	i (A/m^2^)	OCP (mV)	b_a_
1	2.190 ± 0.021	−622 ± 5	5.162 ± 0.043
7	0.023 ± 0.001	−584 ± 4	4.861 ± 0.031
14	0.015 ± 0.0007	−556 ± 4	4.445 ± 0.026
21	0.008 ± 0.0003	−539 ± 3	4.445 ± 0.025

## Data Availability

The original contributions presented in this study are included in the article. Further inquiries can be directed to the corresponding author.

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
