# Peer review of "Preparation and Performance Study of Graphene Oxide Doped Gallate Epoxy Coatings"

_materials, 2025, doi:10.3390/ma18153536_

Round 1
Reviewer 1 Report
Comments and Suggestions for Authors
The article “Preparation and performance study of Graphene Oxide Doped Gallate Epoxy Coatings” by Junhua Liu et al is devoted to obtaining a polymer-based coating doped with graphene oxide and methyl gallate. The resulting coating was tested as an anticorrosive protective coating for a steel base. The article is relevant, but the reviewer has a number of comments and questions (given below).
Not a very good image of graphene oxide in Figure 1 - in the form of a circle. Graphene oxide is a layered material, and such a designation can mislead a reader who does not specialize in carbon materials. It is also unclear what W in Figure 1 means.
Is the proposed spin-coating method scalable for creating a coating on longer structural elements, or for details with complex geometry?
It would be interesting for the reader to see transverse microsections of the obtained samples. This would immediately remove the emerging issues with EDX analysis and allow us to assess the degree of corrosion penetration.
It is still unclear why the anti-corrosion coating was applied to the surface of the layer with rust, and not without it, which would be more logical. In addition, could it not be possible that an autocatalyzed rusting reaction will begin under the protective coating layer, which will not be noticeable. How was it proven that after interaction with rust, the coating does not separate without the necessary adhesion?
The text "GO can improve corrosion resistance of the coating by enhancing electrical conductivity, compactness, and dispersion of the coating" is unproven, and at the level of a hypothesis. Graphene oxide improves anti-corrosion properties due to its layered structure and a large number of oxygen-containing groups. However, it does not have electrical conductivity, but is a dielectric. The word "compactness" requires clarification.
Also, the coating itself cannot be called fully characterized. What is the contribution of each component, is there an interaction between methyl gallate and graphene oxide and the coating base. Is their presence in the system really necessary? The study needs to be supplemented with some comparative data.
Comments on the Quality of English LanguageThe English text requires proofreading. The text contains lexical and punctuation errors.
Reviewer 2 Report
Comments and Suggestions for Authors
Dear authors,
Thank you for your manuscript and the very interesting research you conducted on a modified epoxy anti-rust coating, using methyl gallate as a rust conversion agent, graphene oxide as a conductive functional material, and epoxy resin.
On this occasion, I kindly ask you to mark, arrange and standardize the size of the images in a better way and improve their quality. I also ask that you use the same font and color on all locations and write exponents correctly. Please display the graphical data more clearly in Figure 4c, so that their corrosion current density values are visible for all time intervals.
Given the very close OCP values at different test time intervals, please provide information on the number of repeated tests for each interval as well as their values in order to prove their average value.
Author Response
Thank you very much for taking the time to review this manuscript. Please find the detailed responses below and the corresponding revisions/corrections highlighted/in track changes in the re-submitted files.
Comments 1: On this occasion, I kindly ask you to mark, arrange and standardize the size of the images in a better way and improve their quality. I also ask that you use the same font and color on all locations and write exponents correctly. Please display the graphical data more clearly in Figure 4c, so that their corrosion current density values are visible for all time intervals.
Response 1: Thank you for pointing this out. We agree with this comment. We have arranged and standardized the size of Figures 2,3,4,6 and 9; We have optimised and refined Figures 4, 8. Exponents and font issues were adjusted. Additionally, the values of current density are shown in Figure 4c.
Comments 2: Given the very close OCP values at different test time intervals, please provide information on the number of repeated tests for each interval as well as their values in order to prove their average value.
Response 2: Thank you for pointing this out. We have accordingly provided information on the number of repeated tests, and corresponding content is marked in red.
Response to Comments on the Quality of English Language
Response 1: (in red)
Reviewer 3 Report
Comments and Suggestions for Authors
Dear authors,
The document is quite interesting, and your proposal to use gallate (methyl gallate) as an additive is not just an idea, but a potential breakthrough. Its reactivity and compatibility with epoxy could lead to significant advancements in the field. However, there are some important aspects that need to be addressed:
a) The introduction lacks key recent references on graphene and GO in epoxy coatings.
b) The methodology section should provide a more detailed description of the GO dispersion methodology, as this is a crucial step in the research. For instance, it is currently unclear what ultrasound power and frequency were used, which significantly affects the dispersion process.
c) In the EDS results, it's important to include atomic percentages along with the images and counts. This will provide a more comprehensive understanding of the elemental composition of the surface, which is necessary for the research.
d) There are no SEM/EDS cross-sectional images to demonstrate the actual thickness of the coating (~125 μm), uniformity of coverage, and adhesion to the surface.
e) The interpretation of the SVET data is interesting, but is 3600 s sufficient to conclude stability?
f) EIS mentions a maximum Rct of ~2259 kΩ, but the conclusions refer to 46 kΩ.
g) It is necessary to clarify what the “impedance value” mentioned in the conclusions is.
Comments on the Quality of English Language
As for the wording, it needs to be reworked to make it more fluid, using shorter and clearer paragraphs.
Reviewer 4 Report
Comments and Suggestions for Authors The authors have reported the fabrication of anticorrosion coatings using an epoxy-based formulation on mild steel substrates. The authors claimed that their anti-rust properties are enhanced by the presence of methyl gallate (MG) and graphene oxide(GO). The paper has the merit to be published but it has to be amended by including the necessary control experiments: 1. There is lack of control experiments through all this research. The authors have only compared the properties of the uncoated substrate and the coated substrate (containing Epoxy-MG-GO). However, in order to investigate the role of MG and GO, it is necessary to study the properties of the control coatings: Epoxy, Epoxy-MG and Epoxy-GO. Therefore, I would suggest performing all the experiments for the new control coatings (Epoxy, Epoxy-MG and Epoxy-GO) and compare the results to the properties of the anti-rust coating (Epoxy-MG-GO).2. The quality of the figures 4,6 and 8 needs to be improved.
Reviewer 5 Report
Comments and Suggestions for Authors
Review of the article Preparation and performance study of Graphene Oxide Doped Gallate Epoxy Coatings (Manuscript ID: materials-3738686)
Abstract. Coatings that are tolerant of poor surface preparation are often used for rapid, real-time maintenance of aging steel surfaces. In this study, a modified epoxy (EP) anti-rust coating was proposed by utilizing methyl gallate (MG) as a rust conversion agent, graphene oxide (GO) as conductive functional material, and epoxy resin as the film-forming material. The anti-rust mechanism was investigated through potentiodynamic polarization (PDP), electrochemical impedance spectroscopy (EIS), scanning electron microscopy (SEM), laser scanning confocal microscope (LSCM) and scanning vibration electrode technique (SVET). The results demonstrated that over a period of 21 days, the impedance of the coating increases while the corrosion current density decreases with prolonged soaking time. The coating exhibited a maximum impedance of 46 kΩ, and lower corrosion current density of 8.316×10–3 A/m2, which demonstrated a three-order magnitude reduction compared to the corrosion current density observed in mild steel without coating. LSCM demonstrated that MG can not only penetrate into the tiny gap between the rust particles, but also effectively convert harmful rust into a complex. SVET showed a much more uniform current density distribution in the micro-regions of mild steel with the anti-rust coating compared to uncoated mild steel, indicating that the presence of GO not only enhanced the electrical conductivity of the coating, but also improved the structure of the coating, which contributed to high performance of the modified epoxy anti-rust coating. This work highlights the potential application of anti-rust coating in the field of protection for metal structures in coastal engineering.
1. Novelty
This study presents a compelling investigation into the development of an anti-corrosion coating that incorporates methyl gallate (MG) as a rust conversion agent and graphene oxide (GO) as a functional filler in an epoxy matrix. The combination of these components, particularly in the context of application to poorly prepared steel surfaces, appears to be relatively unexplored and holds promising potential.
The use of MG as an environmentally friendly alternative to conventional rust converters, such as tannic or phosphoric acids, is scientifically relevant. The demonstrated ability of MG to form stable complexes with corrosion products adds to the coating's performance. Furthermore, the inclusion of GO enhances both the conductivity and barrier properties of the system, contributing to improved corrosion resistance.
That said, the manuscript would benefit from a clearer articulation of what differentiates this approach from prior work, particularly coatings based on GO or traditional organic inhibitors.
2. Scope of the Journal
The subject matter of the manuscript aligns well with the scope of Materials, particularly its focus on advanced coatings, corrosion protection technologies, and applied surface engineering. The development of a modified epoxy coating for corrosion-prone steel structures, especially in marine and coastal environments, clearly fits within the journal’s emphasis on material performance and environmental durability.
3. Significance
The results are clearly presented and generally well interpreted. The authors demonstrate that the inclusion of methyl gallate (MG) and graphene oxide (GO) substantially enhances the corrosion resistance of the epoxy coating system. The combination of potentiodynamic polarization (PDP), electrochemical impedance spectroscopy (EIS), scanning vibration electrode technique (SVET), and surface characterization (SEM, LSCM) provides a comprehensive and convincing dataset.
After 21 days of immersion in simulated seawater, the coated samples showed a corrosion current density (icorr) of 8.316 × 10⁻3 A/m2, compared to 25.84 A/m2 for uncoated steel, a reduction of three orders of magnitude. The impedance increased correspondingly, with the charge transfer resistance (Rct) reaching 2259 kΩ·cm2, indicating excellent barrier performance. SVET data confirmed a highly uniform current distribution in the coated sample, with average microzone current densities remaining below 22 μA/cm2, in contrast to over 700 μA/cm2 in the uncoated condition after 1 hour.
These quantitative improvements are significant and strongly support the authors' conclusion that MG contributes to rust conversion through complexation, while GO enhances coating compactness and conductivity. The conclusions are well justified and of clear relevance to corrosion protection in aggressive environments, particularly where surface preparation is limited.
4. Quality of Presentation
The manuscript presents meaningful experimental results; however, in its current form, it requires substantial editorial and technical revision. Despite the scientific value of the study, the clarity, consistency, and overall presentation fall short of the standards expected by Materials, and this significantly impacts readability and perceived rigor.
Already in the introduction (lines 42–44), the text contains a syntactically overloaded sentence regarding the inadequate quality of surface preparation. This type of dense and overly formal construction appears throughout the manuscript and should be simplified to enhance readability. This is particularly important in the opening sections, where the tone and clarity of the paper are established.
There are also specific typographical and formatting issues that must be addressed. For example, line 54 includes a double comma (“additives,,”), and lines 130–131 end with a double period (“...as described..”). In line 397, there is a direct repetition of the phrase “the resistance of the coating surface continued to increase,” which appears twice in succession clearly an editorial oversight that should be corrected.
Numerical data formatting is inconsistent. In Table 1 (starting at line 259), some corrosion current density values are reported in standard decimal format (e.g., “2.190±0.021”), while others use scientific notation (e.g., “(1.547±0.013)×10⁻2”). This inconsistency is visually disruptive and should be corrected by adopting a uniform format throughout the table. Furthermore, the parameter “ba” is listed without explanation, which may confuse readers unfamiliar with electrochemical terminology. All table entries should be accompanied by clearly defined terms and units.
The quality of the graphical materials also needs improvement. In Figure 2(d) (EDS elemental mapping), the contrast is too low for reliable interpretation, and the absence of a scale bar is a serious omission, particularly for SEM images, where spatial reference is essential. More critically, the manuscript lacks cross-sectional SEM images of the coating. Given that the authors claim a coating thickness of up to 150 μm and draw conclusions about its structural integrity and adhesion, surface-only images are insufficient. Cross-sectional micrographs are necessary to assess the uniformity, density, and interface bonding of the coating, and are standard practice in
studies on protective coatings. Without them, the structural claims remain only partially substantiated.
There are also issues with the use of abbreviations. Acronyms such as OCP, icorr, SVET, LSCM, and EIS are introduced inconsistently, often used without prior definition or repeated without clarification. A full list of abbreviations and units should be provided either at the beginning of the manuscript or in a footnote for clarity and consistency.
In summary, while the experimental work is valuable, the manuscript requires a thorough technical and linguistic revision. This should include correction of typographical and punctuation errors, standardization of units and data formats, consistent use of abbreviations, enhanced figure quality, and the inclusion of cross-sectional SEM data. A professional language edit by a native English speaker with experience in scientific writing is also strongly recommended. Only after these revisions are made will the manuscript meet the quality standards expected by Materials.
5. Scientific Soundness
The study is scientifically sound and based on a well-structured experimental design. The combined use of PDP, EIS, SVET, LSCM, SEM/EDS, and XRD provides a comprehensive evaluation of the coating’s performance. Triplicate measurements support data reliability.
The corrosion mechanism is generally well discussed, particularly the role of methyl gallate (MG) in forming stable iron complexes. However, the manuscript would benefit from a deeper mechanistic explanation at the molecular level, especially regarding Fe–MG coordination, which is central to the proposed protective effect. Supporting evidence from spectroscopic techniques (e.g., FTIR, Raman) or reference to prior mechanistic studies could strengthen this aspect.
Additionally, the absence of cross-sectional SEM imaging limits the evaluation of coating thickness and interface quality. More precise details on formulation components (e.g., dispersant and curing agent ratios) and clearer justification for the EIS circuit model, including goodness-of-fit metrics, are also recommended.
6. Interest to the Readers
The manuscript addresses a practical and timely problem corrosion protection on poorly prepared steel surfaces which is highly relevant to readers working in materials engineering, surface protection, and marine infrastructure. The use of MG and GO in a functional epoxy system, combined with thorough electrochemical and structural analysis, offers applied value and scientific interest. While the topic is specialized, it aligns well with the journal’s audience and will be of clear interest to researchers focused on advanced coatings and corrosion-resistant materials.
7. Overall Merit
The manuscript presents a well-executed study with clear practical relevance in the field of corrosion protection. The use of methyl gallate and graphene oxide in an epoxy system is applied thoughtfully to address real-world challenges, particularly on surfaces with minimal preparation. The findings are supported by a solid set of electrochemical and surface analyses, and the observed improvements in corrosion resistance are convincing.
However, despite its strengths, the manuscript requires further refinement before it is suitable for publication. In particular, the absence of cross-sectional SEM imaging limits the structural assessment of the coating, and the discussion of the corrosion inhibition mechanism would benefit from deeper molecular-level insight. These aspects are critical to fully support the conclusions drawn.
In its current form, the work holds strong potential and offers meaningful advancement in applied materials research, but it should be revised to meet the scientific and presentation standards expected by the journal.

The manuscript is generally understandable, and the scientific content can be followed without major difficulty. However, the quality of English falls below the level expected for publication in an international journal. The text contains numerous grammatical errors, awkward phrasing, inconsistent terminology, and typographical issues that impact clarity and fluency.
Several sentences are overly complex or imprecise in structure, requiring rewriting for readability. There are also recurring problems with punctuation, formatting of units, and use of abbreviations. These issues, while not obscuring the core message, do detract from the professionalism and readability of the manuscript.
Round 2
Reviewer 3 Report
Comments and Suggestions for Authors
Dear authors:
It's commendable that most of the critical points have been satisfactorily addressed, leading to a more coherent and complete manuscript.
While the manuscript has improved, it's crucial to address the remaining limitations. The absence of SEM/EDS cross-sectional images, although justified, and the relatively short duration of the SVET test (3600 s) could still be questioned during peer review, though it is not strictly essential if you clearly acknowledge these limitations in the discussion.. Future work should focus on including cross-sectional images or extending the SVET measurements to provide a more comprehensive understanding.
Reviewer 5 Report
Comments and Suggestions for Authors
Thank you for your careful revision of the manuscript. I appreciate your efforts to take into account the reviewers' comments. The current version is much improved in both content and layout.
Regarding the figures, while I understand your decision to remove the problematic EDS images, I would like to emphasize that the inclusion of high-quality cross-sectional SEM images may further strengthen your future studies. Such images will provide more direct evidence to support your conclusions about coating thickness and adhesion properties.
The improvements in language are noticeable and the manuscript now reads much more clearly. The organization of abbreviations and technical terms is now properly done.